Journal of
open psychology data

# The Relation Between the Public Attitude Towards COVID-19 and its Applied Policies – a Dataset for Binational and Temporal Comparison

DATA PAPER

**NOEMI HUBER** (iD)

**RAPHAEL BUCHMÜLLER** (iD)

**ULF-DIETRICH REIPS** (iD)

*Author affiliations can be found in the back matter of this article

]u[ ubiquity press

## ABSTRACT

The here presented data were collected to explore the relationship between people's attitudes toward COVID-19 measures and policy strictness. We conducted online surveys in July 2020 and May 2021 with 131 respectively 130 participants from Switzerland and Germany. Participants responded on visual analogue scales to 33 respectively 25 questions. Further data on participants' information sources, health status, and demographics were collected. The data contribute to understanding psychological and behavioural reactions to COVID-19 policies and may help to further examine the pandemic policy management. The dataset, coding, and variables can be found online on PsychArchives (https://doi.org/10.23668/psycharchives.12899). The study was preregistered on OSF (https://osf.io/uw8mh/).

**CORRESPONDING AUTHOR:**

**Noemi Huber**

University of Konstanz, DE

noemi.huber@uni-konstanz.de

**KEYWORDS:**
COVID-19; corona policies; public attitude; compliance; binational comparison; time-related development

**TO CITE THIS ARTICLE:**
Huber, N., Buchmüller, R., & Reips, U.-D. (2023). The Relation Between the Public Attitude Towards COVID-19 and its Applied Policies – a Dataset for Binational and Temporal Comparison. *Journal of Open Psychology Data,* 11: 12, pp. 1–9. DOI: https://doi. org/10.5334/jopd.84

                                                                                                                 

# (1) BACKGROUND

The COVID-19 pandemic globally triggered a political discourse on state intervention and policies in crisis situations within the governments as well as in its affected populations. To prevent the further spread of the Coronavirus, countries implemented policies while accepting the restriction of an individual's rights and well-being as well as negative effects on economic and social cohesion.

Countries worldwide clearly differ in their prevention strategies regarding their individual measures, strictness and timing (Mathieu et al., 2020). Meanwhile, the applied policies often have found varying degrees of support within their countries' populations (Miguel et al., 2021; Nivette et al., 2021).

To better understand peoples' reactions to national measures for crisis management and democratic conflict resolution, it is critical to collect data to pinpoint and compare public opinion on crisis-related policies (Li et al., 2020). In this publication, we present data on populational attitudes towards a country's COVID-19 measures and its policies' strictness. More precisely, we enable a transnational comparison of the Swiss and German population stances towards the implemented measures on two critical points in time.

The two main questions of the data survey were (1) how public attitudes differ between the Swiss and German populations with respect to its different Corona policies. And (2) how the attitudes of the populations towards national crisis measure change over time.

The first survey was distributed four months after the beginning of the lockdown in Germany and Switzerland. Due to the topics' novelty at that time, no literature existed to describe the populational stance towards the implemented COVID-19 measures. Therefore, this study was conducted as an exploratory approach to capture single snapshots of the public opinion to apprehend the new situation.

Information on the number of cases (Mathieu et al., 2020), the implemented corona policies (Bundesamt für Gesundheit BAG, 2022; Regierung des Landes Baden-Württemberg, 2022) and the measures' relative strictness (Mathieu et al., 2020) can be retrieved to evaluate the mentioned research questions.

# (2) METHOD

## 2.1 STUDY DESIGN

The data was collected via an online questionnaire which was distributed twice among Swiss and German residents. The data should be treated as cross-sectional because the sets of participants in the two surveys might not be identical.

Participants were recruited via WhatsApp (https://www.whatsapp.com) through snowball sampling. They were asked to fill out the survey and pass on the link. The sampling process was initiated through colleagues, friends and students from the University of Konstanz.

The survey's focus was the importance of compliance and appropriateness towards currently implemented policies and a person's fear regarding COVID-19. The questions were compiled to resemble the major impacts of the given corona strategies and can be outlined by the following subtopics: 'importance of mask-wearing' (e.g. I find it important to wear a mask in order to protect myself.), 'importance of restrictions to private meetings' (e.g. In my opinion all contact restrictions for private meetings in public places should be lifted.), 'importance of restrictions to events' (e.g. In my opinion all restrictions for clubs should be lifted.), 'presence of COVID-19 within the individuals' environment' (e.g. I perceive the topic COVID-19 as present in conversations with others.).

For additional analysis, demographic questions were asked (e.g. How old are you? Are you male, female or diverse? Where do you live? Do you have any illness which makes you a risk patient? Do you smoke?).

The second questionnaire was additionally extended by questions on the personal COVID-19 situation (e.g. Did you/your family have COVID-19? Are you vaccinated?).

All variables of the two surveys and their English translation can be found on PsychArchives (https://doi.org/10.23668/psycharchives.12899).

## 2.2 TIME OF DATA COLLECTION

In order to draw a temporal comparison, the data was collected at two specific points in time. The first time of data collection was from the 12th of July to the 14th of August 2020, which according to Mathieu et al. (2020) was about two months after the first wave of COVID-19 incidences. The second time of data collection was from the 29th of April to the 15th of May 2021, which was about half a month after the second and shortly before the third wave of COVID-19 incidences in Germany and Switzerland.

According to Mathieu et al. (2020) during the first time of data collection, there were approximately 10 (12.07.2020) to 23 (14.08.2020) cases per million people in Switzerland while there were 4 (12.07.2020) to 12 (14.08.2020) cases per million people in Germany. During the second time of data collection, there were approximately 226 (29.04.2021) to 146 (15.05.2021) cases per million people in Switzerland and 239 (29.04.2021) to 132 (15.05.2021) cases per million people in Germany (see Table 1).

The measures strictness in Germany and Switzerland at two times of data collection can be compared with the stringency index by Mathieu et al. (2020). The stringency index includes nine different measures: 'school closures, workplace closures, cancellation of

public events, restrictions on public gatherings, closures of public transport, stay-at-home requirements, public information campaigns, restrictions on internal movements, and international travel controls'. It takes a value between 0 and 100, where 100 stands for the strictest measure. At the time of the first survey measures, strictness was between 55.09 and 56.94 in Germany and between 39.35 and 43.06 in Switzerland. At the time of the second survey measures, strictness was higher in both countries, namely 75.00 in Germany and 50.93 in Switzerland (see Table 2).

Measures of the two countries regarding mask-wearing, private meetings and crowds (Bundesamt für Gesundheit BAG, 2022; Regierung des Landes Baden-Württemberg, 2022) were generally more strict at the time of the second survey than at the time of the first one (see Table 3).

## 2.3 LOCATION OF DATA COLLECTION

Data were collected in Germany and Switzerland. At the time of the first survey, 53 participants reported to live in Germany and 74 in Switzerland. At the time of the second

| | DATE | COVID-19 CASES (PER MILLION) | |
| --- | --- | --- | --- |
| | | GERMANY | SWITZERLAND |
| First survey | 12.07.2020 | 4 | 10 |
| | 14.08.2020 | 12 | 23 |
| Second survey | 29.04.2021 | 239 | 226 |
| | 15.05.2021 | 132 | 146 |

**Table 1** COVID-19 Cases per Million People in Germany and Switzerland at the Time of the First and the Second Survey.

*Note*: The number of cases are according to Mathieu et al. (2020).

| | DATE | MEASURES STRICTNESS | |
| --- | --- | --- | --- |
| | | GERMANY | SWITZERLAND |
| First survey | 09.07.2020 | 55.09 | 39.35 |
| | 14.08.2020 | 56.94 | 43.06 |
| Second survey | 28.04.2021 | 75.00 | 50.93 |
| | 15.05.2021 | 75.00 | 50.93 |

**Table 2** Strictness of Measures in Germany and Switzerland at the Time of the First and the Second Survey.

*Note*: Measures strictness according to the stringency index by Mathieu et al. (2020). The stringency index includes 9 different measures ("school closures; workplace closures; cancellation of public events; restrictions on public gatherings; closures of public transport; stay-at-home requirements; public information campaigns; restrictions on internal movements; and international travel controls") and takes a value between 0 and 100 (100 = strictest measures).

survey, 67 participants reported to live in Germany and 62 in Switzerland. Most participants reportedly came from the border region of the two countries, so from the federal state of Baden-Wuerttemberg (Germany) and the canton Thurgau (Switzerland).

## 2.4 SAMPLING, SAMPLE AND DATA COLLECTION

In the first data collection, 157 participants opened the survey and 135 of them read and agreed to the informed consent, then stated that they want to seriously participate in the survey ("seriousness check" technique, see Reips, 2021). Of these participants, four were not included in the analysis because they stated to be younger than 18 years old. The data of the potentially underaged participants (four who stated to participate seriously and 3 who stated to not participate seriously) were removed from the data set due to privacy regulations. Out of the 131 remaining 54 participants lived in Germany (21 male, 32 female, 1 no answer) and 74 lived in Switzerland (30 male, 44 female). Three participants dropped out of the survey before answering the question. The German participants on average were 32 years old (SD 13.69). The Swiss participants had a mean age of 36 years (SD 16.23).

In the German sample, 8 participants out of 50 identified themselves as a smoker (41 non-smokers, 1 no answer). Also, 8 German participants stated to suffer from a disease that makes them a high-risk patient (37 no such disease, 5 not sure).

In the Swiss sample, 14 participants out of 70 identified themselves as a smoker (55 non-smokers, 1 no answer). Seven Swiss participants stated to suffer from a disease that makes them a high-risk patient (61 no such disease, 2 not sure).

In the second data collection, 138 participants opened the survey and 133 of them read and agreed to the informed consent, then confirmed that they would like to seriously participate. Again, three participants were not included in the analysis because they stated to be younger than 18 years old. The data of the potentially underaged participants were removed due to privacy regulations. Of the 130 remaining participants, 67 lived in Germany (18 male, 47 female, 1 diverse, 1 no answer) and 63 lived in Switzerland (36 male, 27 female). The German participants on average were 27 years old (SD 10.60). The Swiss participants had a mean age of 40 years (SD 17.88).

In the German sample, 6 participants out of 61 identified themselves as a smoker (57 non-smokers, 1 no answer). Also, 5 German participants stated to suffer from a disease which makes them a high-risk patient (54 no such disease, 1 not sure, 4 no answer).

In the Swiss sample, 13 participants out of 56 identified themselves as a smoker (43 non-smokers).

| | | MEASURES | |
|---|---|---|---|
| | | GERMANY | SWITZERLAND |
| First survey | Mask-wearing | -in shopping centers and stores<br>-in public transportation, at train- and bus-stations<br>-at the doctors<br>-in cosmetic facilities<br>-for workers in public entertainment and restaurants | -in public transportation |
| | Private Meetings | -gatherings of no more than 20 people | No measures |
| | Crowds | -limited to 100 people<br>-Dances forbidden<br>-Clubs and discos closed | -Discos, dance halls and clubs limited to 1000 people<br>-if more than 300 people: there has to be a separation to sectors<br>-when the distance of 1,5m cannot be held: the contact details have to be collected |
| Second survey | Mask-wearing | -in shopping centers and stores<br>-in public transportation<br>-in closed public rooms<br>-at workplaces and schools<br>-at religious events<br>-services with close body contact<br>-in the car with more than one household<br>-if incidence over 100: mandatory to wear FFP2/KN95/K95-masks | -in public places (e.g. restaurants, museums)<br>-in public transportation<br>-at workplaces and schools<br>-in closed public rooms<br>-at sport-, cultural- or other events |
| | Private Meetings | -maximum of 5 people in public and private places<br>-incidence 100: 1 household plus 1 other person is allowed<br>-incidence under 35: meetings with 10 people from 3 households | -maximum of 15 people (with some further restrictions) |
| | Crowds | -gatherings and events forbidden with some exceptions (court trials, exams, etc.) | -events with audience possible up to 100 people outside and 50 people inside (with some further restrictions) |

**Table 3** Some COVID-19-related Measures in Germany and Switzerland at the Time of the First and the Second Survey.

*Note*: The measures are drawn from the Bundesamt für Gesundheit BAG (2022) and Regierung des Landes Baden-Württemberg (2022).

7 Swiss participants stated to suffer from a disease which makes them a high-risk patient (45 no such disease, 2 not sure, 2 no answer).

In the second survey, participants additionally were asked about their personal COVID-19 situation. The questions were whether they have/had COVID-19, whether someone in their household had it and whether someone in their family (siblings, grandparents, nieces, etc.) had been infected. In the German sample, 3 out of 65 participants answered to have/have had COVID-19, 2 answered that someone in their household had been infected and 18 answered that someone in their extended family had been infected. In the Swiss, sample 8 out of 61 participants answered to have/have had COVID-19, 6 answered that someone in their household has had it and 32 answered that someone in their extended family had been infected.

In the second survey participants were also asked whether they are vaccinated against COVID-19. In the German sample, 4 out of 65 participants stated to be fully vaccinated and 14 people stated to be partly vaccinated. In the Swiss sample, 6 out of 61 participants answered

to be fully vaccinated and 5 participants answered to be partly vaccinated.

During both times of data collection, participants did not receive any payment for their participation in the survey. In the first survey, participants could write down an e-mail address to receive information about the analysed data.

## 2.5 MATERIALS/SURVEY INSTRUMENTS

The data was collected at two distinct points in time with two slightly different versions of the online questionnaires, for exact wording see repository information below. The questionnaire was created and conducted with the platform SoSci Survey (Leiner, 2019).

The first survey contained 31 questions + seriousness check (4 demographic questions + 8 questions about the general attitude toward COVID-19 + 5 mask wearing questions + 4 questions about private meetings during the pandemic + 3 questions about the limitation of big crowds + 5 questions about the communication of the COVID-19 situation + 1 question about smoking and +1 question about pre-existing illnesses). The questions in

the first survey were presented in this order. The second survey contained 24 questions + seriousness check (4 demographic questions + 1 question about the personal COVID-19 situation + 6 questions about the general attitude toward COVID-19 + 4 mask-wearing questions + 1 question about private meetings during the pandemic + 1 question about the limitation of big crowds + 5 questions about the communication of the COVID-19 situation + 1 question about smoking and + 1 question about pre-existing illnesses). The questions in the second survey were presented in this order. It took about 10 minutes to answer all the questions in the survey. A progress bar was shown on the upper right side to inform the participants about the current progress.

During the seriousness check, participants were asked whether they want to seriously participate in the study or just want to click through. Only data from serious participants were included in the analysis.

Participants were asked about their sex, age, place of residence (Germany or Switzerland), their federal state or canton, whether they consider themself to be a smoker and whether they suffer from one of the diseases listed (adiposity degree III, high blood pressure, chronic respiratory problem, diabetes, cardiovascular disease, cancer, chronic disease of the liver). Participants were not allowed to continue the survey without answering all demographic questions, because these questions were considered to be important for later analysis. The participants were asked to answer every question, but also had the option to click on "I do not wish to provide any further information on this page."

All questions about the general attitude towards COVID-19, mask-wearing, private meetings, limitation of big crowds and 4 out of 5 questions about the communication of the COVID-19 situation were measured on a visual analogue scale (VAS) with the endpoints "not true at all" and "completely true" (original endpoints in German were "trifft überhaupt nicht zu" and "trifft voll und ganz zu"). While responses on the VAS were coded from 1 to 101, no numerals were visible to the respondents. One question about sources of information and questions about the personal COVID-19 situation were multiple-choice questions.

In response to evolving research objectives and to reduce the burden on survey participants, we made strategic modifications to our questionnaire by removing questions that elicited primarily neutral responses, on account of their perceived lack of specificity. For example, we removed PT03_01, which probed the respondents' perceptions regarding the efficacy of limiting interpersonal contact in public spaces as a measure for controlling the spread of the virus. Given the potential for varied and ambiguous interpretations of the responses to this item, we concluded that its inclusion was unlikely to yield meaningful insights. Some non-specific questions were also reformulated in the second

survey, to make them clearer. The majority of questions remained the same to enable the comparison of results. A table with an overview of all items used and changes in item formulation can be found in the appendix (see Table 4).

In the second wave of the survey, we also added one question about the personal COVID-19 situation, because we felt it was important to consider the influence of previous experience with the COVID-19 situation.

*Informed consent*: In the informed consent, participants were told prior to the compiled questions that the goal of this survey was to compare the risk perception and attitude towards COVID-19 measures of German and Swiss inhabitants. They were also informed that participation is voluntary, that it takes approximately 15 minutes (in the first survey)/ 5–10 minutes (in the second survey), that the data is anonymous and only used for scientific reasons and that it is always possible to interrupt or terminate the survey. They were asked to read the questions carefully. It was pointed out that everyone can participate, who is at least 18 years old and has a domicile in Germany or Switzerland. In the first survey it was also mentioned, that at the end of the survey, participants can enter an e-mail address to receive the results of the study. In the second survey participants were also informed that they were allowed to participate in this survey when they had already participated in the first survey. The informed consent of both times of data collection can be found in the appendix.

The complete dataset, variables, questions and coding of the two survey waves can be found online on PsychArchives (https://doi.org/10.23668/psycharchives.12899).

## 2.6 QUALITY CONTROL

Before data collection, the survey was pilot tested with three people. They were asked to fill out the survey and give feedback. After pilot testing, minor changes in spelling and wording were made.

Participants were asked at the beginning of the survey whether they wanted to seriously participate (seriousness-check) as a measure of quality control.

To promote complete data, participants were reminded to answer every question whenever they left one unanswered. However, participants then had the option to click on "I do not wish to provide any further information on this page".

## 2.7 DATA ANONYMISATION AND ETHICAL ISSUES

Under German and Swiss law, no ethics board approval was needed for the data collection. The survey was anonymous. At the end of the first survey, participants had the possibility to write down an e-mail address to receive information about final results of the study. All potential identifiers like e-mail addresses and information

about the federal state or canton as well as data from underaged participants were removed from the data set. At the beginning of the study, participants were informed about the anonymity and asked whether they would like to seriously participate.

### 2.8 EXISTING USE OF DATA
No publications or outputs have originated from this data.

## (3) DATASET DESCRIPTION AND ACCESS

### 3.1 REPOSITORY LOCATION
Huber, N., Buchmüller, R. & Reips, U.-D. (2021). Dataset for: The relation between the Public Attitude towards COVID-19 and its Applied Policies – A Binational and Temporal Comparison [Data set]. PsychArchives. https://doi.org/10.23668/psycharchives.12899.

### 3.2 OBJECT/FILE NAME
data_first_survey.CSV
data_second_survey.csv
encoding_first_survey.csv
encoding_second_survey.CSV
variables_first_survey.CSV
variables_second_survey.CSV

### 3.3 DATA TYPE
data_first_survey.CSV: in this file, you can find the data of the first survey (executed in July and August 2020)

data_second_survey.csv: in this file, you can find the data of the second survey (executed in April and May 2021)

encoding_first_survey.csv: in this file, you can find the coding of the answer options of the first survey

encoding_second_survey.CSV: in this file, you can find the coding of the answer options of the second survey

variables_first_survey.CSV: in this file, you can find the questions and answer options of the first survey

variables_second_survey.CSV: in this file, you can find the questions and answer options of the second survey

### 3.4 FORMAT NAMES AND VERSIONS
All data files are saved as CSV. The data files are available on PsychArchives in version 4.

### 3.5 LANGUAGE
The data files are stored in German and American English (translated).

### 3.6 LICENSE
The data has been deposited under a Scientific Use License (v1).

### 3.7 LIMITS TO SHARING
Data is available for scientific use only. The data have no time restriction.

### 3.8 PUBLICATION DATE
The data set was first published on 16/10/2021. It was embargoed until 01/07/2022. The current version of the dataset (version 4) was published on 30/05/2023 and now is open for scientific use without time restriction.

### 3.9 FAIR DATA/CODEBOOK
Findability: The data is stored in PsychArchives of the Leibniz Institute (https://www.psycharchives.org). It is described and can easily be found and identified by its DOI, title, and authors.

Accessibility: The data can be easily retrieved by its identifier and is available for scientific use without time restriction.

Interoperability: The data is provided in English and German.

Reuse: The variables and the encoding necessary to use the data is uploaded in PsychArchives. The data is described in this paper. There is information about the usage license and the provenance.

## (4) REUSE POTENTIAL

At the time of survey creation, the pandemic had just recently fully reached European countries and thus only little research about the impact of the virus and policy response existed at that time. Therefore, this survey must be understood as an experimental approach and further research is crucial for more explicit conclusions. Nonetheless, the data from this survey can highlight possible areas of interest which offer the potential for further research. (e.g. differences in attitudes between German and Swiss population).

Many types of analyses on this data set might be interesting for other researchers. It can be used to analyse the correlation between demographic details and the attitude and awareness towards the actual policy management. For example, there might be a correlation between people who are vaccinated and their attitude towards mask-wearing. Or there might be a correlation between the risk status (high-risk vs. low-risk patient) and how informed a participant feels. Further research is possible towards the relation between information flow and public attitudes.

The data generation followed major quality control principles. Participants were asked at the beginning of the survey whether they want to seriously participate in it (seriousness-check). This check improves data quality because it reduces participant drop-out during the study (Reips, 2021). The questionnaire is built in a way that most pages only contain one question (One Item One

Screen; OIOS). This design enables researchers to analyse the response times and dropouts (Reips, 2010). Most answers in the survey were given on a visual analogue scale (VAS) from 1 to 101, which allows the collection of detailed responses. Another strength of the survey is, that most constructs (e.g. attitude towards mask-wearing) are measured with various questions. Therefore, topics are treated more thoroughly. The participant sample consists of a wide age range (18 to 80 years) which promotes representativeness.

An individual characteristic of the data is the critical points in time of its collection. Both data collection methods happened at comparably low incidence rates between disease waves. The first data collection was about four months after the lockdown in Germany and Switzerland. The second data collection was one year later, right before the third wave. Both points in time were discussed to be critical regarding the insecurity towards the further development of the COVID-19 virus and the enforcement or relaxation of national policies.

Limitations are, that the sample size in the data set is rather small (survey 1: N = 143, survey 2: N = 133) and participants of the Swiss sample share had a higher mean age on average in both times of surveys. The recruitment was performed through snowball sampling instead of random selection (Parker, 2019). This method was chosen due to its convenience to reach participants during the pandemic. The snowball sampling process was initiated through colleagues, friends and students of the University of Konstanz. Due to the small sample size and the recruitment through snowball sampling the question of representativity of the data for the Swiss and German populations can be raised. This concern is reasonable. The study followed an explorative approach and its data should be treated accordingly.

Another limitation of this study is that the first and the second versions of the questionnaire slightly differ. The second questionnaire contains fewer questions than the first one and some questions are formulated differently in the second questionnaire. The reason for the adaption of questions was to reduce the survey burden on the participants and ask some questions more specific. This change in questions limits the comparability of the results of the two surveys. However, the majority of questions stayed consistent to enable the comparison.

Participants were not compensated for their participation in this survey. This is not unusual for a short survey of 10 minutes. But it may result in low participation rates and non-response of people who are not interested in the topic. However, not compensating participants can also have positive effects like high-quality responses, because participants are intrinsically motivated to take part in the survey.

The data touches multiple fields of research including politics, sociology and health research and therefore can be utilized collaboratively. To evaluate the results, they should be cross-validated with results from other studies that are being published currently during the post-pandemic phase (see e.g. our very recent paper by Shevchenko, Huber & Reips, 2023, in which a different methodology was used). Further analyses of the data can contribute to a better understanding of psychological and behavioural reactions to COVID-19 policies and can help to further improve pandemic policy management. The investigation and exploration of the data may generate new hypotheses regarding opinion mining of COVID-19 research. Furthermore, the data can be used in combination with similar data sources to gain further knowledge on the topic of public perspectives towards policy management. For instance, Meier et al. (2020) investigated similar results. A combination of data could prove to be beneficial and help prepare humanity for the next pandemic.

## SPECIAL COLLECTION

This submission is part of the special collection of data papers related to psychological research on the impact of COVID-19 pandemic in the Journal of Open Psychology Data (JOPD). Editors of the Special Collection are Katarina Blask, Rainer Mauer, Martin Kerwer, Alexander Jedinger and Débora B. Maehler.

## ADDITIONAL FILE

The additional file for this article can be found as follows:

- **Appendix.** In the appendix demographic information about the participants' place of residence in the two surveys is displayed. DOI: https://doi.org/10.5334/jopd.84.s1

## FUNDING INFORMATION

Funding was provided by the Chair of Research Methods, Assessment, and iScience, Department of Psychology, University of Konstanz.

## COMPETING INTERESTS

The authors have no competing interests to declare.

## AUTHOR CONTRIBUTIONS

Noemi Huber was primarily responsible for planning, constructing and distributing the survey. She was also in charge of analyzing the data and writing the

one-page proposal as well as this data paper. Raphael Buchmüller acted as active support during the planning and constructing of the survey. He revised the one-page proposal and the manuscript. He especially helped a lot with the planning of the survey and writing this paper. Prof. Dr. Ulf-Dietrich Reips supervised, provided feedback, advised, and revised proposal, materials, and manuscript.

## AUTHOR AFFILIATIONS

**Noemi Huber** orcid.org/0000-0002-2258-4321
University of Konstanz, DE

**Raphael Buchmüller** orcid.org/0000-0002-0612-8828
University of Konstanz, DE

**Ulf-Dietrich Reips** orcid.org/0000-0002-1566-4745
University of Konstanz, DE

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

## PEER REVIEW COMMENTS

*Journal of Open Psychology Data* has blind peer review, which is unblinded upon article acceptance. The editorial history of this article can be downloaded here:

- **PR File 1.** Peer Review History. DOI: https://doi.org/10.5334/jopd.84.pr1

Huber et al. *Journal of Open Psychology Data* DOI: 10.5334/jopd.84 **9**

**TO CITE THIS ARTICLE:**
Huber, N., Buchmüller, R., & Reips, U.-D. (2023). The Relation Between the Public Attitude Towards COVID-19 and its Applied Policies – a Dataset for Binational and Temporal Comparison. *Journal of Open Psychology Data,* 11: 12, pp. 1–9. DOI: https://doi.org/10.5334/jopd.84

