## [Peer Review History. · Journal of Open Psychology Data]

Dear Noemi Huber, Raphael Buchmüller, Ulf-Dietrich Reips,

After review, we have reached a decision regarding your submission to Journal of Open Psychology Data, "The Relation between the Public Attitude towards COVID-19 and its Applied Policies - A Dataset for Binational and Temporal Comparison". Our decision is to request revisions of the manuscript prior to acceptance for publication.

The full review information is included at the bottom of this email. Please note that there may also be a copy of the manuscript file with reviewer comments available once you have accessed the submission account. We ask you to please consider the following issues and revise the file accordingly:

Major revisions:

As both reviewers mentioned, there are serious ethical concerns with your data. In particular, the reviewers and I wondered why there was no ethical review for the project presented. In addition, participants were not asked for their informed consent. However, in one of your published datasets, there are some direct identifiers (email addresses) and data from underage participants was also published. I know from your paper that the email addresses were given voluntarily by the participants. However, in accordance with EU-GDPR I would like to ask you to remove the direct identifiers and also the data of underage participants from your data (as recommended by reviewer 2) and publish this data as a new version. In addition, I would ask you to explain exactly why you did not see the need for an ethics vote to be obtained.

Minor revisions:

Please provide more information on the snowball sampling method and carefully check again the data quality and documentation of your data (for detailed information see comments from both reviewers).

Instructions for how to resubmit your article online are pasted below. Please ensure that your revised files adhere to our author guidelines, and that the files are fully proofed prior to upload. Please also include a revised version of your article with 'tracked changes', adding comments where appropriate, to indicate the revisions made, in addition to a brief document outlining how you have responded to the reviewers' requests.

If you have trouble processing the revisions, our Help Center (<https://help.u-community.io>) or downloadable PDF (<https://bit.ly/Author-Guide-OJS-3>) may be able to help. If not, please get in touch and we'll be happy to help.

Please also ensure that all copyright permissions have been attained for any figures/tables you have included.

Please could you have the revisions submitted with two weeks. If you cannot make this deadline, please let us know as early as possible.

Kind regards,

Katarina Blask

You must upload your revised files and the response as follows:

1) login to the journal account with your username and password

2) access 'My Queue' and click 'View' for the submission in question

3) click 'Upload File', within the Revisions section

4) (a pop-up window will display) Select the article component you are uploading (e.g. manuscript, figure etc) and then either drag your file into the displayed area or click 'Upload file' to select it from your files. When done click 'Continue'

5) check the file name is appropriate (edit it here if not) and then click 'Continue'

6) if you wish to upload another revised file, click 'Add another file', otherwise, click 'Complete'

If you need to upload more files, repeat steps 3-6

Reviewer 1:

I would like to thank authors for the opportunity to review your paper "The relation between the Public Attitude towards COVID-19 and its Applied Policies - A Binational and Temporal Comparison". Overall, I find the study to be well-conducted and informative, with several strengths including the questions to measure attitudes, health status and the inclusion of demographic information from both Germany and Switzerland in 2020 and 2021. However, I have some major concerns regarding the ethical implications of your study and the privacy and confidentiality of your participants.

First and foremost, I have some concerns about the relatively small sample size, particularly in the second survey, and the use of snowball sampling. While snowball sampling can be a useful method for accessing populations during the COVID-19 pandemic, it may introduce bias into the results as participants are recruited through personal networks. It would be helpful if the authors could provide a more detailed description of how they initiated the snowball sampling process and whether they had any initial contacts to avoid bias. Additionally, the authors should address this limitation in the discussion and consider strategies for increasing sample size and improving representativeness in future studies.

Second, while the questionnaire was piloted and minor changes were made, I noticed that some questions were reformulated between the two surveys, which may limit the ability to compare results over time. I suggest the authors provide a more detailed explanation for why changes were made and how this may affect interpretation of the results.

Moving on to the next point, and the most importantly, I found that one of the data files ("data_first_survey.csv") contains a list of participants' email addresses. I can see 67 emails and can (google) search them online to get more information about them, such as the name, the company they work etc. This is highly concerning as it raises questions about the privacy and confidentiality of the participants. It is important that you take immediate action to remove this file from the public repository and ensure that participants' personal information is not made publicly available in the future.

As data protection regulations are becoming increasingly strict, it is important to consider whether participants would have wanted their data to be publicly available indefinitely. Particularly, once data is published with a DOI, it is generally not possible to remove it from the repository where it was published, as this would compromise the integrity and reproducibility of the research. However, there may be some exceptional circumstances where data can be removed, such as in cases of legal or ethical violations, or if the data contains sensitive information that was not adequately protected.

Fourthly, to share data publicly, is important to ensure that the data is accurate and reliable. In the "data_first_survey" file, column "DF02_01" which refers to age, has some inconsistencies. For example, case 71 has a value of "Durchschnittlich 17" while others have numerical values. It is important to clarify what this value means and how it ended up in the dataset. Similarly, there are cases where the age is listed as "79 Jahre". It would be helpful if the authors could explain how they cleaned and validated the data to ensure accuracy and reliability. Additionally, the file "varibales_first_survey" contains some question marks, it isn't clear what is the difference between this file to "encoding_*" files.

The fifth point to be discussed is no mention of compensation for participants in the paper. It would be helpful if the authors could provide more information on whether participants were compensated for their time and participation in the study. If participants were compensated, it is important to clarify the nature and amount of compensation provided. If not, it would be useful to discuss the potential impact this could have on participation and data quality.

Finally, I appreciate the value of your study and the potential insights it can offer on the public perception of COVID-19 policies. However, it is concerning that no ethical approval was obtained for data collection. As per standard practice, ethical approval should be sought for any research involving human subjects. This is especially important for research on sensitive topics such as the COVID-19 pandemic.

Reviewer 2:

The submission by Huber et al. describes the data of two online surveys conducted at different times during the Covid-19 pandemic with German and Swiss samples.

The method section provides sufficient detail to recreate the survey with the help of the provided datasets. However, it appears that there are no indications which scale endpoints the visual analog scales used (neither in the text nor in the data). It is also unclear if cases excluded from analysis (e.g., underage individuals) were also removed from the datasets (I still found cases aged 17 or younger).

The reuse section provides reasonable suggestions for future research and acknowledges the limitations due to the small sample size and non-representative sampling.

The data itself are published in an open format and are labeled in a way that is easy to understand. The only information missing appears to be the labeling of VAS endpoints (see above).

I would recommend stating explicitly whether the order presented in the dataset corresponds to the order of questions in the survey.

Finally, I must note severe ethical concerns. The study was not reviewed (which is likely fine under institutional guidelines), but the dataset for survey 1 includes email addresses, some with identifiable names. The data thus allow for direct identification of participants (different from what is suggested in the paper). The authors need to remove this information immediately or clarify that participants were explicitly consenting to have their email addresses published in an openly-accessible repository. Similarly, data from underage participants should not be published, given these participants were not able to consent to participate in the study in the first place.

Dear Noemi Huber, Raphael Buchmüller, Ulf-Dietrich Reips,

Thank you for submitting a revised version of your manuscript "The Relation between the Public Attitude toward COVID-19 and its Applied Policies - A Dataset for Binational and Temporal Comparison" and for your comments on the reviewers' feedback. We have now reviewed all of the changes you made and thank you for your efforts to address the reviewers' feedback.

Nevertheless, in light of the ethical and legal concerns expressed by both reviewers, I would have actually expected a more thorough review of your data as well as the relevant paragraphs in the manuscript. In particular, I would have expected a thorough review for critical indirect and direct identifiers in the data, i.e., variables that could contribute to re-identification of participants in addition to the direct identifier "email address". For example, by specifying the federal state/canton for your participants, you have another critical indirect identifier in your data. This is mainly because almost all participants, at least from the German subsample, come from Baden-Württemberg. Therefore, the information on the federal state or canton of the participants should be anonymized, i.e., you should coarsen it both in the sample description and in the data itself. Within the data, you can decide whether to delete the corresponding variable completely or to use rather broad categories, such as 1 = Baden Württemberg, 2 = St. Gallen, 3 = Thurgau, 4 = other federal state, 5 = other canton. To get more information about possible anonymization strategies, you can also have a look at the following document: https://www.pedocs.de/frontdoor.php?source_opus=21970. In this context, I would also like to ask you to delete Table 4.

In order to more transparently reflect the ethical standards applied in your study, I would also like to ask you to include the consent form used as an appendix.

Finally, it would be nice if the explanations requested by the reviewers regarding changes in wording or use of items from time 1 to 2 could be made clearer by including a corresponding table in the appendix.

Instructions for how to resubmit your article online are pasted below. Please ensure that your revised files adhere to our author guidelines, and that the files are fully proofed prior to upload. Please also include a revised version of your article with 'tracked changes', adding comments where appropriate, to indicate the revisions made, in addition to a brief document outlining how you have responded to the requests.

If you have trouble processing the revisions, our Help Center (<https://help.u-community.io>) or downloadable PDF (<https://bit.ly/Author-Guide-OJS-3>) may be able to help. If not, please get in touch and we'll be happy to help.

Please also ensure that all copyright permissions have been attained for any figures/tables you have included.

Please could you have the revisions submitted with two weeks. If you cannot make this deadline, please let us know as early as possible.

Kind regards,

Katarina Blask

You must upload your revised files and the response as follows:

- 1) login to the journal account with your username and password
- 2) access 'My Queue' and click 'View' for the submission in question
- 3) click 'Upload File', within the Revisions section
- 4) (a pop-up window will display) Select the article component you are uploading (e.g. manuscript, figure etc) and then either drag your file into the displayed area or click 'Upload file' to select it from your files. When done click 'Continue'
- 5) check the file name is appropriate (edit it here if not) and then click 'Continue'
- 6) if you wish to upload another revised file, click 'Add another file', otherwise, click 'Complete'

If you need to upload more files, repeat steps 3-6.

Dear Noemi Huber, Raphael Buchmüller, Ulf-Dietrich Reips,

After review, we have reached a decision regarding your submission to Journal of Open Psychology Data, "The Relation between the Public Attitude towards COVID-19 and its Applied Policies - A Dataset for Binational and Temporal Comparison", and are happy to accept your submission for publication.

As just discussed in the video meeting, please make the following minor revisions:

Please, release the data under a Scientific Use License in accordance with the information provided in the consent form. Along with this, you should also adjust the DOI and the versioning information in your manuscript.

Instructions for how to resubmit your article online are pasted below. Please ensure that your revised files adhere to our author guidelines, and that the files are fully proofed prior to upload. Please also include a revised version of your article with 'tracked changes', adding comments where appropriate, to indicate the revisions made, in addition to a brief document outlining how you have responded to the reviewers' requests.

If you have trouble processing the revisions, our Help Center (<https://help.u-community.io>) or downloadable PDF (<https://bit.ly/Author-Guide-OJS-3>) may be able to help. If not, please get in touch and we'll be happy to help.

We will contact you again with any final revisions prior to publication - this will also be your final chance to proofread your manuscript and check for any final issues prior to publication.

Kind regards,

Katarina Blask

You must upload your revised files as follows:

- 1) login to the journal account with your username and password
- 2) access 'My Queue' and click 'View' for the submission in question
- 3) click 'Upload File', within the Revisions section
- 4) (a pop-up window will display) Select the article component you are uploading (e.g. manuscript, figure etc) and then either drag your file into the displayed area or click 'Upload file' to select it from your files. When done click 'Continue'
- 5) check the file name is appropriate (edit it here if not) and then click 'Continue'
- 6) if you wish to upload another revised file, click 'Add another file', otherwise, click 'Complete'

If you need to upload more files, repeat steps 3-6.